# Adaptation of hydroxymethylbutenyl diphosphate reductase enables volatile isoprenoid production

Mareike Bongers[1,2]*, Jordi Perez-Gil[2,3], Mark P Hodson[2,4,5], Lars Schrübbers[1], Tune Wulff[1], Morten OA Sommer[1], Lars K Nielsen[1,2†], Claudia E Vickers[2,6†]*

[1]Novo Nordisk Foundation Center for Biosustainability, Technical University of Denmark, Lyngby, Denmark; [2]Australian Institute for Bioengineering and Nanotechnology, The University of Queensland, Brisbane, Australia; [3]Centre for Research in Agricultural Genomics (CRAG) CSIC-IRTA-UAB-UB, Campus UAB Bellaterra, Barcelona, Spain; [4]Metabolomics Australia, Australian Institute for Bioengineering and Nanotechnology, The University of Queensland, Brisbane, Australia; [5]School of Pharmacy, The University of Queensland, Brisbane, Australia; [6]CSIRO Synthetic Biology Future Science Platform, Brisbane, Australia

*For correspondence:
marbon@biosustain.dtu.dk (MB);
c.vickers@uq.edu.au (CEV)

†These authors contributed equally to this work

**Abstract** Volatile isoprenoids produced by plants are emitted in vast quantities into the atmosphere, with substantial effects on global carbon cycling. Yet, the molecular mechanisms regulating the balance between volatile and non-volatile isoprenoid production remain unknown. Isoprenoids are synthesised via sequential condensation of isopentenyl pyrophosphate (IPP) to dimethylallyl pyrophosphate (DMAPP), with volatile isoprenoids containing fewer isopentenyl subunits. The DMAPP:IPP ratio could affect the balance between volatile and non-volatile isoprenoids, but the plastidic DMAPP:IPP ratio is generally believed to be similar across different species. Here we demonstrate that the ratio of DMAPP:IPP produced by hydroxymethylbutenyl diphosphate reductase (HDR/IspH), the final step of the plastidic isoprenoid production pathway, is not fixed. Instead, this ratio varies greatly across HDRs from phylogenetically distinct plants, correlating with isoprenoid production patterns. Our findings suggest that adaptation of HDR plays a previously unrecognised role in determining in vivo carbon availability for isoprenoid emissions, directly shaping global biosphere-atmosphere interactions.

## Introduction

Biogenic volatile organic compounds (BVOCs) emitted from the biosphere have significant effects on global climate and air quality (*Loreto and Fares, 2013*). Short-chain isoprenoids such as isoprene, a $C_5$ hydrocarbon, contribute more than 80% of BVOCs, totalling about 650 million tonnes of carbon per year (*Sindelarova et al., 2014*). The vast quantity and high reactivity of emitted volatile isoprenoids affect the oxidative capacity of the troposphere (*Thompson, 1992*; *Wennberg et al., 2018*), impact the residence time of the greenhouse gas methane (*Fehsenfeld et al., 1992*), and contribute to air pollution through formation of secondary organic aerosols, surface-level ozone and carbon monoxide (*Claeys et al., 2004*; *Poisson et al., 2000*; *Granier et al., 2000*; *Figure 1*). The effects of isoprenoid emissions may be exacerbated by climate change and shifts in land use (*Peñuelas and Staudt, 2010*), warranting a better understanding of how plants accomplish and regulate these vast emissions.

All isoprenoids are made from the $C_5$ isomers isopentenyl pyrophosphate (IPP) and dimethylallyl pyrophosphate (DMAPP) (*Figure 1*). Two non-homologous metabolic pathways produce DMAPP and IPP in plants: the cytosolic mevalonic acid (MVA) and the plastidic methylerythritol phosphate

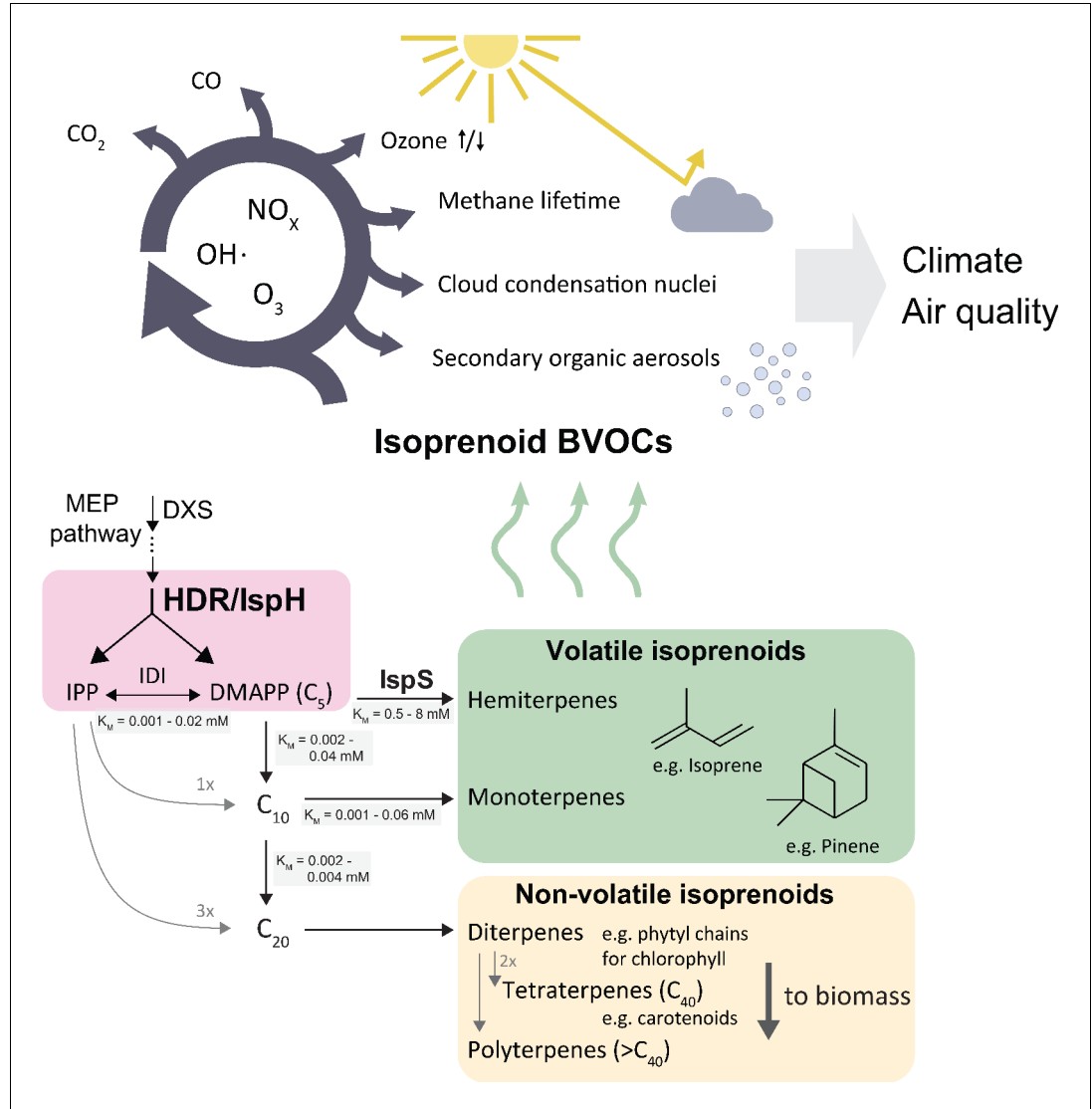

**Figure 1.** Simplified scheme of the plastidic MEP pathway, important volatile isoprenoids, and their atmospheric reactions. The MEP pathway makes IPP and DMAPP simultaneously through the action of HDR (pink box), and produces the bulk of volatile isoprenoids, contributing >80 % of total BVOCs (*Sindelarova et al., 2014*) . Non-volatile isoprenoids are essential and synthesised by all organisms, while volatile isoprenoid production is non-essential and highly species-dependent. The cytosolic MVA pathway contributes most sesquiterpenes (<3 % of BVOCs), but is omitted here for clarity. Emitted volatile isoprenoids are rapidly oxidised, resulting in complex atmospheric photochemistry impacting aerosol and cloud condensation nuclei formation, extension of methane residence time, ozonolysis as well as surface-level ozone formation in the presence of mono-nitrogen oxide ($NO_x$) pollutants (*Wennberg et al., 2018*). BVOCs, biogenic organic volatile compounds; DMAPP, dimethylallyl pyrophosphate; DXS, deoxyxylulose synthase; IDI, isopentenyl diphosphate isomerase; IPP, isopentenyl pyrophosphate; IspS, isoprene synthase; HDR, hydroxymethylbutenyl diphosphate reductase.

(MEP) pathways, the latter contributing almost all volatile isoprenoids (*Pulido et al., 2012*). The final step of the MEP pathway is catalyzed by the enzyme hydroxymethylbutenyl diphosphate reductase (HDR/IspH), which produces both IPP and DMAPP (*Figure 1*). Isoprenoid chain length is initially determined by how many units of IPP are condensed with one molecule of DMAPP, before terpene synthases and other modifying enzymes convert these intermediates into isoprenoids. The resulting compounds are classified by carbon chain length.

In plants, longer-chain isoprenoids ($C_{15}$ and higher) serve many essential roles, e.g. as membrane components and parts of the photosynthetic apparatus (*Pulido et al., 2012*; *Figure 1*). Short-chain

isoprenoids ($C_5$, $C_{10}$, and some $C_{15}$ compounds) are volatile under physiological conditions, and their functions are generally not essential for plant survival (*Vickers et al., 2009*). It is currently unknown how plants control carbon allocation between short-chain and long-chain isoprenoids in the chloroplast. While the demand for essential isoprenoids (for example, photosynthetic pigments) is assumed to be relatively similar across plants (*Monson et al., 2013*), different species produce markedly different amounts of non-essential, short-chain volatile isoprenoids (*Wiedinmyer et al., 2020*). For example, some oak (*Quercus*) species produce vast amounts of isoprene, while closely related oaks produce little or none at all (*Wiedinmyer et al., 2020*). Synthesis of isoprenoids with different chain lengths requires different DMAPP:IPP substrate ratios. Much more IPP than DMAPP is needed for long-chain isoprenoid production, so presumably high relative IPP concentrations are necessary for chain elongation while an excess of DMAPP and insufficient IPP could favour short-chain isoprenoid production. Isoprene synthase (IspS) uses only DMAPP, but not IPP, as a substrate.

Volatile isoprenoid emissions can represent a significant loss of carbon; for example, up to 20% of recently fixed carbon can be emitted as isoprene in high-emitting plants (*Sharkey and Loreto, 1993*). Isoprene synthase (IspS) has a high $K_m$ for its substrate DMAPP (0.5–8 mM; *BRENDA, 2020*); despite this, it successfully competes with prenyl phosphate synthases, which typically have $K_{M (DMAPP)}$ values 10- to 100-fold lower (*BRENDA, 2020*). Similarly, monoterpene synthases, which also show lower affinity for the substrates (*BRENDA, 2020*), compete with downstream prenyl phosphate synthases. Hence, the relative abundance of DMAPP may determine the balance between volatile and non-volatile isoprenoids.

Here we examined HDR as a potential mechanism to provide variability in the DMAPP:IPP ratio. Previous studies in diverse organisms (*Escherichia coli*, the bacterium *Aquifex aeolicus*, red pepper chromoplasts, and cultured tobacco cells) all found that HDR produces DMAPP:IPP ratios between 1:4 and 1:6 (*Rohdich et al., 2003*; *Altincicek et al., 2002*; *Adam et al., 2002*; *Tritsch et al., 2010*). Consequently, it has been assumed that HDR has a fixed product ratio of about 1:5. However, none of these species produce significant amounts of volatile isoprenoids (*Wiedinmyer et al., 2020*). Isopentenyl diphosphate isomerase (IDI) interconverts DMAPP and IPP, but the reaction is slow (*Jonnalagadda et al., 2012*) and IDI is rate-limiting for isoprenoid production generally, including isoprene (*Vickers et al., 2014*). We hypothesised that HDR enzymes from species that emit large amounts of short-chain volatile isoprenoids produce a higher ratio of DMAPP to IPP, which could support production of volatiles like isoprene.

## Results and discussion

We selected *HDR* genes from the bacterium *E. coli*, *Synechococcus sp.* strain PCC 7002 (a photosynthetic prokaryote) and eight species from diverse taxa of the plant kingdom (*Table 1*). Many plants harbour more than one annotated *HDR* gene, some of which may be pseudogenes. Therefore, we first identified functional HDR genes by their ability to complement an otherwise lethal knockout of the *ispH/HDR* gene in *E. coli* (*Altincicek et al., 2001*). We found at least one functional gene from each species (*Figure 2—figure supplement 1a*); however, severe dose-dependent growth defects were observed when overexpressing certain *HDR* genes, possibly due to toxicity of prenyl phosphates (*George et al., 2018*; *Figure 2—figure supplement 1b*). This precluded accurate steady-state metabolite quantification and required alleviating toxicity by the introduction of a metabolic sink for IPP and DMAPP. Here we used a lycopene ($C_{40}$ isoprenoid) biosynthetic pathway, including expression of a heterologous *idi* (*Cunningham et al., 1994*). Deoxyxylulose synthase (DXS), the primary rate-limiting step of the MEP pathway, was also overexpressed in order to achieve intracellular IPP and DMAPP concentrations above quantification limits in *E. coli*.

A spectrum of DMAPP:IPP ratios was observed, ranging from almost exclusive IPP production (*Picea sitchensis* HDR1) to almost exclusive DMAPP production (*Populus trichocarpa and Ricinus communis*, *Figure 2a*). A control without *HDR* overexpression (labelled (-) in *Figure 2a*) showed a DMAPP:IPP ratio of ~1.5 to 1 in our experimental setup, serving as a reference point. Overexpressing the *E. coli* HDR shifted the ratio slightly towards IPP, in agreement with previous reports (*Rohdich et al., 2002*). However, HDR enzymes from species known to emit volatile isoprenoids produced considerably more DMAPP - a noteworthy exception being *P. sitchensis* HDR1 (PsHDR1, *Figure 2a*).

**Table 1.** Genetic information and volatile isoprenoid emission profiles for species studied in this work.

Key: blank cell indicates species has not been tested, or genome sequence (or other information) not available; Y indicates significant emissions of isoprene or isoprenoids have been detected, or gene/transcript has been identified; N indicates significant emissions of isoprene or isoprenoids have NOT been detected, or gene/transcript has NOT been identified; MTs, monoterpenes; IspS, isoprene synthase; TPS, terpene synthase.

| Kingdom | Phylum/Clade | Clade | Genus, species | Common Name | HDR protein accession number | E. coli construct Genbank ID | Complements?† | Emissions Isoprene (C5) | Emissions MTs (C10) | Gene/transcript* IspS | Gene/transcript* Short chain TPS | Reference |
|---|---|---|---|---|---|---|---|---|---|---|---|---|
| Plantae | Angiosperms | Eudicots | Ricinus communis | castor bean plant | XP_002519102.1 | MH605331 | yes | N | Y | N | Y | Wiedinmyer et al., 2020; Kadri et al., 2011; Xie et al., 2012 |
| Plantae | Angiosperms | Eudicots | Populus trichocarpa‡ | black cottonwood | 1 ACD70402 | MH605329 | yes | Y | Y | Y | Y | Wiedinmyer et al., 2020; Tuskan, 2006 |
|  |  |  |  |  | 2 PNT41333.1 | MH605330 | no |  |  | —— |  |  |
| Plantae | Angiosperms | Eudicots | Prunus persica | peach | XP_007199828.1 | MH605326 | yes | N | Y | N | Y | Wiedinmyer et al., 2020; Verde et al., 2013 |
| Plantae | Angiosperms | Eudicots | Eucalyptus grandis | flooded gum | 1 XP_010028563.1 | MH605323 | yes | Y | Y | Y | Y | Wiedinmyer et al., 2020; Myburg et al., 2014 |
|  |  |  |  |  | 2 XP_010047332.1 | MH605324 | no |  |  |  |  |  |
| Plantae | Angiosperms | Eudicots | Theobroma cacao | cacao tree | XP_007042717.1 | MH605333 | yes | N | Y | N | Y | Wiedinmyer et al., 2020; Argout et al., 2008 |
| Plantae | Angiosperms | Eudicots | Arabidopsis thaliana | thale cress | AEE86362.1 | MH605322 | yes | N | Y | N | Y | Sharkey et al., 2005; Chen et al., 2004; Bohlmann et al., 2000 |
| Plantae | Angiosperms | Monocots | Elaeis guineensis | oil palm | XP_010909277.1 | MH605325 | yes | Y | Y |  | Y | Wiedinmyer et al., 2020; Wilkinson et al., 2006 |
| Plantae | Gymnosperms | Pinophyta | Picea sitchensis | Sitka spruce | 1 ACN40284.1 | MH605327 | yes | Y | Y |  | Y | Wiedinmyer et al., 2020; Hayward et al., 2004 |
|  |  |  |  |  | 2 ACN39959.1 | MH605328 | yes – toxic |  |  |  |  |  |
| Bacteria | Cyanobacteria |  | Synechococcus sp. PCC 7002 | Synechococcus | ACA98524.1 | MH605332 | yes | N | N | N | N |  |

* Identified from data/genomes available on NCBI (https://www.ncbi.nlm.nih.gov/) and literature search (references noted).

† Whether protein expression was able to functionally complement an E. coli ΔispH knockout in this study.

‡ Also known as Populus balsamifera ssp. trichocarpa.

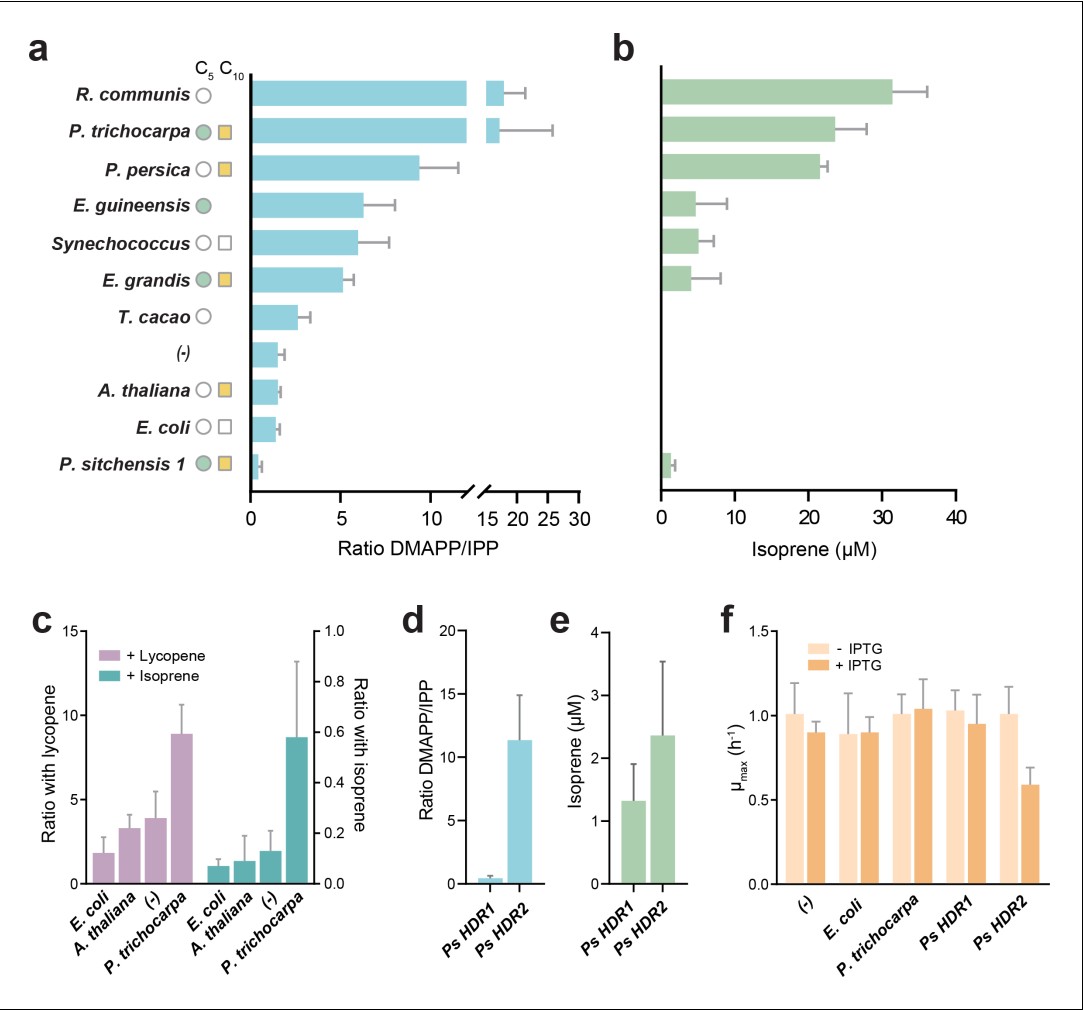

**Figure 2.** DMAPP:IPP ratio and isoprene production with different HDR enzymes. (**a**) In vivo ratio of DMAPP:IPP measured via LC-MS/MS in *E. coli* overexpressing *HDR* genes from different species, in the genetic context of *dxs* and lycopene biosynthetic pathway overexpression. Filled circles and squares indicate that the HDR source species natively emits $C_5$ or $C_{10}$ isoprenoids. Open symbols indicate no emission, and no symbol indicates no data or conflicting data. (**b**) Isoprene production in *E. coli* when the HDR enzymes shown in panel (a) are overexpressed with *dxs* and an isoprene synthase. (**c**) Comparison of DMAPP:IPP ratios between selected *HDR*s co-expressed with *dxs* and with expression of either lycopene or isoprene as the metabolic sink. (**d**) Comparison of DMAPP:IPP ratios in *E. coli* overexpressing *Picea sitchensis* (*Ps*) *HDR1* or *HDR2* in the context of *dxs* and lycopene biosynthetic pathway overexpression. (**e**) Isoprene production in *E. coli* overexpressing *P. sitchensis HDR1* or *HDR2* along with *dxs* and an isoprene synthase. (**f**) The maximum specific growth rate ($\mu_{max}$) of *E. coli* expressing selected *HDR*s in the context of *dxs* and lycopene biosynthetic pathway overexpression, with or without induction of *HDR* expression by addition of IPTG. All data shown as mean ± SD from ≥ 3 biological replicates; (-) indicates the control strain without *HDR* overexpression.

The online version of this article includes the following source data and figure supplement(s) for figure 2:

**Source data 1.** Raw data for metabolomics, proteomics and isoprene measurements shown in *Figure 2* and supplements.

**Figure supplement 1.** Complementation of lethal knockout of *ispH* in *E. coli* using different *HDR*s, and associated DMAPP toxicity.

**Figure supplement 2.** Protein quantification of IDI, HDR and the lycopene biosynthetic pathway.

**Figure supplement 3.** Quantification of DMAPP and IPP using LC-MS/MS.

These values do not represent direct product ratios of the examined HDRs due to the presence of the heterologously expressed lycopene pathway and *idi*. However, they show that product ratios vary up to 40-fold between HDRs, and that the assumed fixed 1:5 DMAPP to IPP ratio is in fact an exception, rather than the rule. Using LC-MS proteomics, we tested whether the observed phenotypes were influenced by differences in expression of the native *E. coli* HDR, IDI, or the plasmid-encoded lycopene production pathway. We found no difference in protein levels in any of the HDR overexpression strains compared to the no HDR overexpression control (one-way ANOVA, $p>0.05$), except for the anticipated increase in *E. coli* HDR in the respective overexpression strain (Welch's ANOVA, Dunnett's *post hoc* test $p<0.005$; *Figure 2—figure supplement 2*). Because no shared

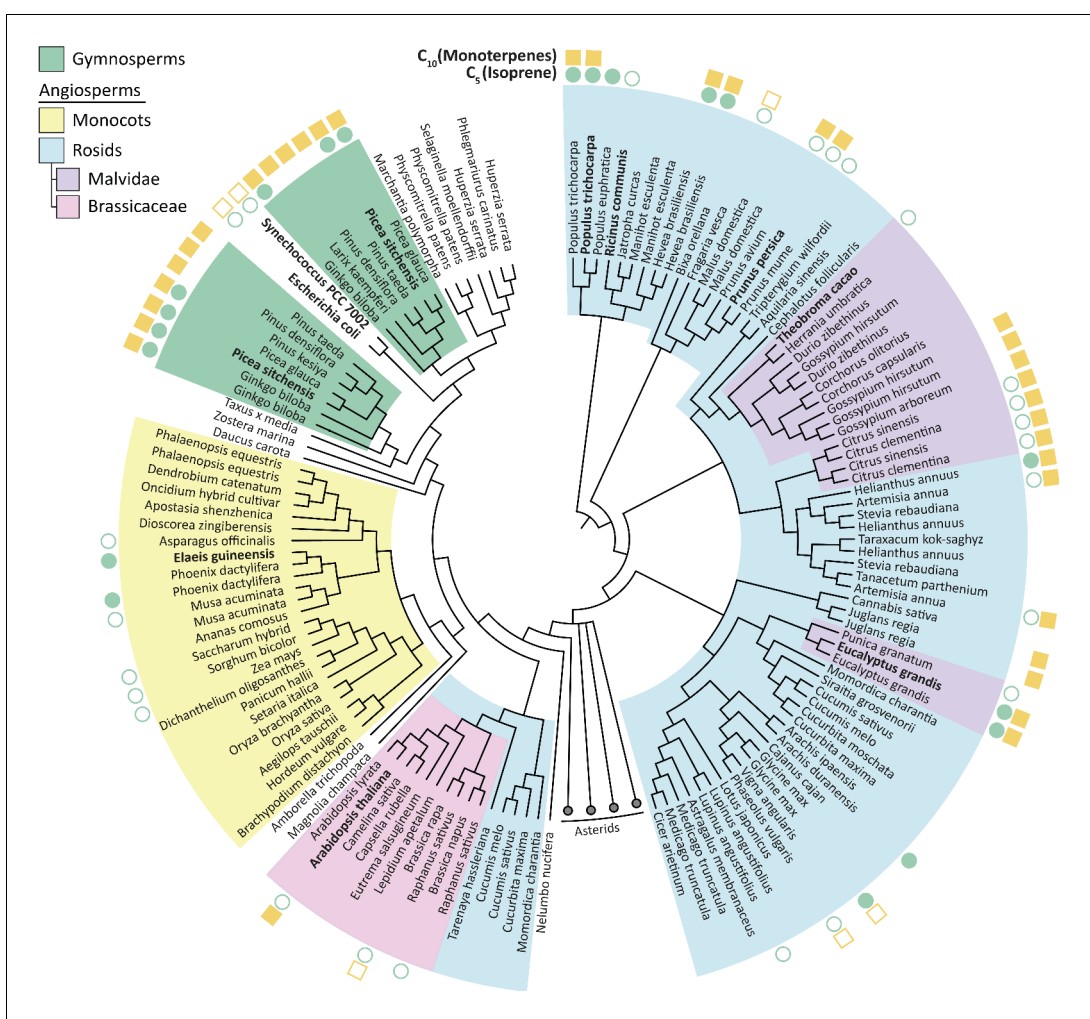

**Figure 3.** Phylogenetic tree of HDR proteins from land plants, the cyanobacterium *Synechococcus* and *Escherichia coli*. Where known, each species' $C_5$ (isoprene) and $C_{10}$ (monoterpenes) emission spectra are shown (*Wiedinmyer et al., 2020*). High DMAPP-producing HDR proteins (from *P. trichocarpa*, *R. communis* and *P. persica*) cluster together based on high sequence similarity. Homologues within species, such as *P. trichocarpa*, tend to be highly similar; except for in gymnosperms where two separate groups of likely paralogous HDRs exist. Proteins analysed in this study are highlighted in bold. The Asterids clade is collapsed for clarity. Tree generated from BLAST sequence alignment with *A. thaliana* HDR against all land plants, using maximum likelihood phylogeny. Empty symbol, no volatile emission; filled symbol, volatile emission; no symbol, no or conflicting data available.

The online version of this article includes the following source data for figure 3:

**Source data 1.** List of HDR sequences used for phylogenetic analysis in *Figure 3*.
**Source data 2.** Phylogenetic tree of HDRs in Newick format.

proteotypic peptides exist across all heterologous HDRs, quantitative comparison of HDR protein levels across strains is not possible. However, we confirmed that all tested HDRs were strongly over-expressed (*Figure 2—figure supplement 2c*), and that there was no correlation between HDR abundance and the DMAPP:IPP ratio ($rho = -0.488$, data not shown). Taken together, these data demonstrate that different HDR enzymes produce vastly different DMAPP:IPP ratios, with some plant HDRs producing a ratio significantly shifted towards more DMAPP than previously recognized.

To test whether an increased in vivo DMAPP:IPP ratio would favour isoprene production, we replaced the lycopene pathway with an overexpressed isoprene synthase (IspS) as a metabolic sink. A high DMAPP:IPP ratio was indeed closely associated with isoprene production (*Figure 2b*). To confirm that differences in DMAPP:IPP ratios are robust when changing from lycopene ($C_{40}$) to isoprene ($C_5$) production, we compared selected HDR product ratios with both downstream metabolic sinks (*Figure 2c*). While the absolute values shifted towards DMAPP (left y-axis; lycopene requires 6 IPP and 2 DMAPP) or IPP (right y-axis; isoprene is made only from DMAPP) depending on downstream requirements, the relative difference between HDRs remained similar, demonstrating that our experimental setup captures representative differences between the enzymes.

Isoprene was not produced in the presence of *Theobroma cacao*, *Arabidopsis thaliana* or *E. coli* HDR (all species that do not emit short-chain isoprenoids), presumably because the available DMAPP was insufficient for IspS to compete with downstream enzymes (*Figure 2b*). All HDRs from isoprenoid-emitting species enabled isoprene production, supporting our hypothesis. Interestingly, a high DMAPP:IPP ratio and high isoprene production was also observed with HDRs from *P. persica* and *R. communis*, species that emit some monoterpenes but not isoprene (*Wiedinmyer et al., 2020*; *Kadri et al., 2011*). *Pp*HDR and *Rc*HDR have high (>87%) sequence identity with HDR proteins from high isoprene-emitting species *P. trichocarpa* and *Hevea brasiliensis*, respectively (*Figure 3*), but *R. communis* and *P. persica* do not have an isoprene synthase (*Table 1*).

Together, these data suggest that HDR from different plant species has adapted to produce differing ratios of DMAPP to IPP, and that an increased DMAPP:IPP ratio is an important prerequisite for production of isoprene and perhaps other non-essential, short-chain isoprenoids. Our data indicate that a high DMAPP:IPP ratio is a necessary, but not a sufficient requirement for volatile isoprenoid emission. This places HDR at a key junction in the evolution of isoprene emission, a trait that appeared and disappeared several times across the plant kingdom (*Dani et al., 2014*).

*Picea sitchensis* (Sitka spruce) is a coniferous gymnosperm that emits both isoprene and monoterpenes (*Hayward et al., 2004*), but contrary to our expectation *Ps*HDR1 produced the highest relative amount of IPP and showed very low isoprene production in *E. coli* (*Figure 2a and b*). Recently, the HDR from another gymnosperm, *Ginkgo biloba* (*Gb*HDR1), was shown to produce an even lower DMAPP to IPP ratio in vitro (*Shin et al., 2017*). Most sequenced gymnosperms have two or more HDR isoforms which fall into two distinct classes based on sequence similarity (*Kim et al., 2008*; *Figure 3*). Interestingly, transcriptional studies (*Celedon et al., 2017*; *Kim et al., 2009*) suggest that gymnosperm Type II *HDRs* are particularly abundant at the site of monoterpene-rich resin formation and are generally expressed at higher levels than Type I *HDRs* (*Celedon et al., 2017*) (such as *PsHDR1* and *GbHDR1*). It was therefore tempting to speculate that HDR adaptation in gymnosperms has resulted in paralogues with complementary functions: Type I HDRs, which primarily produce IPP, show basal expression throughout the plant, and are important for long-chain isoprenoid production; and Type II HDRs, which primarily produce DMAPP and are expressed where short-chain isoprenoids are made. This prompted us to investigate the Type II HDR from *P. sitchensis* (*Figure 2d–f*).

*Ps*HDR2 failed in our initial complementation assay (data not shown), most likely due to toxicity as no metabolic sink was present for IPP/DMAPP. Indeed, even in the presence of a sink, overexpression of *Ps*HDR2 reduced *E. coli* growth rate by about 50% (*Figure 2f*), a level of toxicity exceeding that of other high DMAPP-producing HDRs. Interestingly, *Ps*HDR2 produced a > 10 fold excess of DMAPP over IPP, while *Ps*HDR1 had a ratio shifted towards more IPP (DMAPP:IPP = 0.447 +/- 0.19; *Figure 2d*). *Ps*HDR2 also enabled higher isoprene production than *Ps*HDR1 (*Figure 2e*), albeit at a lower yield than the other high DMAPP-producing enzymes, which is most likely an effect of the high toxicity in *E. coli*. The complementary product ratios of *Ps*HDR1 and *Ps*HDR2 strongly suggest functional specialization of these genes, making them paralogues in *P. sitchensis*.

While many plants encode more than one *HDR* gene (*Figure 3*), these homologues are often closely related and thus likely arose from relatively recent large-scale genome duplications

(*Saladié et al., 2014*). In gymnosperms, the two HDR homologues are phylogenetically more distant (*Figure 3*) and likely define functionally specialised paralogues. Hence, we propose that two different strategies might have been employed to adapt HDR to isoprenoid production spectra: either using a single HDR and shifting the DMAPP:IPP ratio to allow production of specific isoprenoid profiles (*Figure 2a*), or having two functionally distinct HDRs each dedicated to the synthesis of one isomer (*Figure 2d*). Whether adaptation of HDR is a result of a change in the demand for DMAPP, or whether it is a driver of its release as isoprene and other volatile isoprenoids, is a fascinating question that remains to be answered.

The discovery of HDR enzymes with different product ratios has important implications for heterologous production of industrially valuable isoprenoids such as biofuels, fragrances and pharmaceuticals (*Vickers, 2015*) in engineered microorganisms. We have shown that only certain HDR enzymes enable production of isoprene in our engineered *E. coli*, and our data indicate that the choice of HDR is important to ensure availability of DMAPP and IPP at appropriate relative concentrations to achieve balanced pathway flux towards the product of interest and to avoid DMAPP toxicity. The presented LC-MS/MS method for separation and absolute quantification of the two isomers (*Figure 2—figure supplement 3*) proved crucial for our discovery, and will enable a deeper understanding of the processes regulating isoprenoid biosynthesis in nature and biotechnology.

Demands from downstream metabolism may determine IPP and DMAPP requirements, and could form an evolutionary driver for enzymatic activities that impact their ratio. Our data suggest that the adaptation of HDR to generate different DMAPP:IPP ratios allows for the production of large amounts of short-chain isoprenoids in certain species or tissues. Our findings illuminate the molecular mechanism underlying how plants emit isoprene and suggest a central role for HDR in determining the spectrum of isoprenoids produced by plants, including isoprenoid BVOCs. Unravelling the mechanism by which plants distribute carbon between volatile and non-volatile isoprenoids will help resolve the complex interplay between BVOC emissions, land-use management and climate change.

# Materials and methods

**Key resources table**

| Reagent type (species) or resource | Designation | Source or reference | Identifiers | Additional information |
|---|---|---|---|---|
| Gene (*Escherichia coli*) | *ispH/HDR* | NCBI 'Gene' | Gene_ID:944777; EcoGene:EG11081; ECK0030; lytB | hydroxymethylbutenyl diphosphate reductase |
| Strain, strain background (*Escherichia coli*) | *Escherichia coli* W | ATCC | ATCC:9637 | obtained from L. Nielsen lab, Australia |
| Genetic reagent (*Escherichia coli*) | *E. coli* WΔ*cscR*, lacZ::PtDXS, arsB::PaISPS | This paper and PMID: 21782859 (*Arifin et al., 2011*) | | knockout of *cscR*, knock-in of *PtDXS* and *PaISPS* |
| Genetic reagent (*Escherichia coli*) | *E. coli* WΔ*cscR*, lacZ::MVA, ΔispH | This paper and PMID: 11115399 (*Campos et al., 2001*) | | knock-in of MVA pathway, knockout of *ispH* |
| Genetic reagent (*Populus trichocarpa*) | DXS | NCBI 'Reference Sequence' | XP_006378082.1 | Deoxyxylulose phosphate synthase, gene was truncated for expression in *E. coli* |
| Genetic reagent (*Populus alba*) | ISPS(del2-52, A3T,L70R,S288C) | Patent WO2012058494 (*Beck et al., 2011*) | | Isoprene synthase (Genbank:EF638224) variant, truncated and mutated |
| Recombinant DNA reagent | pLacZ-KIKO (cm) plasmid | PMID: 23799955 (*Sabri et al., 2013*) | Addgene:46764 | used to integrate *PtDXS* into the genome |

*Continued on next page*

*Continued*

| Reagent type (species) or resource | Designation | Source or reference | Identifiers | Additional information |
|---|---|---|---|---|
| Recombinant DNA reagent | pArsBKIKO (cm) plasmid | PMID: 23799955 (*Sabri et al., 2013*) | Addgene:46763 | used to integrate *PaISPS* into the genome |
| Recombinant DNA reagent | pT-HDR plasmids | This paper | derived from pTrc99a | all HDR genes were cloned into this expression vector |
| Recombinant DNA reagent | pAC-LYC04 | PMID: 7919981 (*Cunningham et al., 1994*) | | |
| Recombinant DNA reagent | *Ricinus communis* HDR expression plasmid | Genbank | MH605331 | HDR protein XP_002519102.1 |
| Recombinant DNA reagent | *Populus trichocarpa* HDR 1 expression plasmid | Genbank | MH605329 | HDR protein ACD70402 |
| Recombinant DNA reagent | *Populus trichocarpa* HDR 2 expression plasmid | Genbank | MH605330 | HDR protein PNT41333.1 |
| Recombinant DNA reagent | *Prunus persica* HDR expression plasmid | Genbank | MH605326 | HDR protein XP_007199828.1 |
| Recombinant DNA reagent | *Eucalyptus grandis* HDR 1 expression plasmid | Genbank | MH605323 | HDR protein XP_010028563.1 |
| Recombinant DNA reagent | *Eucalyptus grandis* HDR 2 expression plasmid | Genbank | MH605324 | HDR protein XP_010047332.1 |
| Recombinant DNA reagent | *Theobroma cacao* HDR expression plasmid | Genbank | MH605333 | HDR protein XP_007042717.1 |
| Recombinant DNA reagent | *Arabidopsis thaliana* HDR expression plasmid | Genbank | MH605322 | HDR protein AEE86362.1 |
| Recombinant DNA reagent | *Elaeis guineensis* HDR expression plasmid | Genbank | MH605325 | HDR protein XP_010909277.1 |
| Recombinant DNA reagent | *Picea sitchensis* HDR 1 expression plasmid | Genbank | MH605327 | HDR protein ACN40284.1 |
| Recombinant DNA reagent | *Picea sitchensis* HDR 2 expression plasmid | Genbank | MH605328 | HDR protein ACN39959.1 |
| Recombinant DNA reagent | *Synechococcus sp. PCC 7002* HDR expression plasmid | Genbank | MH605332 | HDR protein ACA98524.1 |
| Commercial assay or kit | Astec Cyclobond I2000 chiral HPLC column | Sigma Aldrich | 20024AST | HPLC column used for IPP/DMAPP separation |
| Chemical compound, drug | Isoprene | Sigma Aldrich | Cat. # I19551 | |
| Chemical compound, drug | Isopentenyl pyrophosphate | Sigma Aldrich | Cat. # I0503 | |
| Chemical compound, drug | Dimethylallyl pyrophosphate | Sigma Aldrich | Cat. # D4287 | |
| Chemical compound, drug | (±)-Mevalonic acid 5-phosphate | Sigma Aldrich | Cat. # 79849 | |

*Continued on next page*

Continued

| Reagent type (species) or resource | Designation | Source or reference | Identifiers | Additional information |
|---|---|---|---|---|
| Chemical compound, drug | Mevalonolactone | Sigma Aldrich | Cat. # M4667 | |
| Software, algorithm | CLC Main Workbench | Qiagen | RRID:SCR_000354 | |
| Software, algorithm | iTOL | PMID: 27095192 (*Letunic and Bork, 2016*) | https://itol.embl.de/ | Interactive Tree of Life |

## Chemicals and reagents

Isoprene (Cat. No I19551), IPP (Ca. No I0503), DMAPP (Ca. No D4287), Isopropyl β-D-thiogalacto-side (IPTG, Cat. No I6758), (±)-Mevalonic acid 5-phosphate (MVA-P, Ca. No 79849) were purchased from Sigma Aldrich. Mevalonate (MVA) was prepared from (±)-mevalonolactone (Sigma Aldrich, Cat. No M4667) through base-catalyzed hydrolysis (*Campos et al., 2001*). Ammonium acetate was purchased from Sigma Aldrich (Ca. No 73594–25 G-F). Acetonitrile hypergrade for LC-MS LiChrosolv (Ca. No 1000292500) and Methanol hypergrade for LC-MS LiChrosolv (Ca. No 1060352500) was purchased from Merck Millipore. Milli-Q water was generated via a Merck Millipore Integral 3 water purification system.

## Gene, plasmid and *E. coli* strain construction

*E. coli* Top10 (Cat. No C404050, Thermo Fischer Scientific) was used for cloning. For all other experiments, *E. coli* W (ATCC 9637) with a knock-out in the *csc* operon (*E. coli* WΔ*cscR* *Arifin et al., 2011*) was used. Plant HDR chloroplast targeting peptides were predicted using the ChloroP 1.1 server (http://www.cbs.dtu.dk/services/ChloroP/). Genes were truncated to remove chloroplast targeting peptides, codon-optimised for *E. coli* (http://idtdna.com/CodonOpt) and synthesised by Integrated DNA Technologies (Singapore). All plant genes were placed under control of the IPTG-inducible *trc* promoter in a pTrc99-derived (*Amann et al., 1988*) vector, generating the pT-HDR series of plasmids. The *DXS* gene from *Populus trichocarpa* (Genbank Accession No. XP_006378082.1) was integrated into the genome using the pLacZ-KIKO(cm) vector (*Sabri et al., 2013*). The chloramphenicol resistance gene was removed from the genome using pCP20 (*Datsenko and Wanner, 2000*). The resulting strain (*E. coli* WΔ*cscR, lacZ::PtDXS*) was transformed with each of the pT-HDR plasmids and pAC-LYC04 (*Cunningham et al., 1994*) for IPP and DMAPP measurements. For isoprene production experiments, an engineered *ISPS* gene from *Populus alba* (Genbank Accession No. EF638224) was integrated into the genome of *E. coli* WΔ*cscR, lacZ::PtDXS* using pArsBKIKO(cm). Apart from removal of the chloroplast-targeting sequence, this gene was also engineered to contain three mutations to enhance specific activity: ISPS(del2-52,A3T,L70R,S288C) (*Beck et al., 2011*).

## Bacterial growth media

LB medium contained 10 g/L tryptone, 5 g/L yeast extract and 10 g/L NaCl. TB medium contained 12 g/L tryptone, 24 g/L yeast extract, 0.4% (v/v) glycerol, 2 mM $MgSO_4$, 1 mM thiamine, 17 mM $KH_2PO_4$, 7.2 mM $K_2HPO_4$. Where indicated, media were supplemented with 1 mM mevalonate and 1 mM L-arabinose for induction of the MVA pathway operon, or with 0.2% (w/v) glucose or 0.1 mM IPTG for repression or induction of the *trc* promoter. All cultures were grown at 37°C with 250 rpm shaking unless stated otherwise.

## Complementation of the ispH/HDR knockout mutant in *E. coli*

A partial MVA pathway under control of the arabinose-inducible $P_{BAD}$ promoter (*Campos et al., 2001*) was cloned into a pLacZ-KIKO(cm) vector and integrated into the *E. coli* WΔ*cscR* genome. This strain (WΔ*cscR, lacZ::MVA*) was used to knock out *ispH* using recombineering (*Datsenko and Wanner, 2000*), making growth dependent on supplementation with mevalonate and arabinose. Each pT-HDR plasmid was transformed into this strain and tested for its ability to grow in the absence of mevalonate and arabinose.

## Growth rate measurements

Cells were grown in LB medium; glucose, mevalonate or IPTG were added where indicated. Precultures were grown at 37°C with 250 rpm shaking in 96-well plates (Corning, Cat No. CLS3799) until stationary phase. Cultures were diluted to a starting optical density ($OD_{600}$) of 0.05 and the growth was monitored in a microplate reader (BioTek ELx808) at 37°C with 700 rpm double-orbital shaking, measuring $OD_{600}$ every 10 min. All bacterial cultures for quantification of specific growth rates, metabolites and isoprene were grown at least in biological triplicates (from 3 single colonies of the same strain), and means +/- standard deviations are shown.

## Fermentations for metabolite measurements

Strains harbouring the different pT-HDR plasmids and pAC-LYC04 were grown for determination of IPP and DMAPP concentrations. Chloramphenicol ($30 \ mg \ L^{-1}$) and ampicillin ($250 \ mg \ L^{-1}$) were added to the media for plasmid maintenance. Precultures were grown in LB medium as described above. A culture volume of 10 ml of TB medium was inoculated with an overnight preculture in 100 ml baffled flasks to a starting $OD_{600}$ of 0.05. Protein expression was induced with 0.1 mM IPTG at an $OD_{600}$ of 0.5. When an $OD_{600}$ of 5 was reached (exponential growth phase in TB medium), cultures were harvested for metabolite quantification.

## Quantification of IPP and DMAPP

Intracellular metabolites were quenched and extracted using a method adapted from *Bongers et al. (2015)*. To harvest, the equivalent of 1 ml of culture of an optical density of $OD_{600}$ = 5 was centrifuged at 4°C for 20 s at 13,000 x *g*, the supernatant was discarded and the pellet snap-frozen in liquid nitrogen. The pellet was resuspended in 95 µl of 90% acetonitrile (v/v) in water and metabolites were extracted by vortexing for 10 min at room temperature. Cell debris was removed by centrifugation at 4°C for 15 min at 13,000 x *g*. Extracts were transferred into HPLC vials, 5 µl internal standard (MVA-P) was added at a final concentration of 16 µM for analysis using liquid chromatography tandem mass spectrometry (LC-MS/MS).

LC-MS/MS data were acquired on an Advance UHPLC system (Bruker Daltonics, Fremont, CA, USA) equipped with a binary pump, degasser and PAL HTC-xt autosampler (CTC Analytics AG, Switzerland) coupled to an EVOQ Elite triple quadrupole mass spectrometer (Bruker Daltonics, Fremont, CA, USA). Separation of the structural isomers IPP and DMAPP was achieved by adapting a method from *Köhling et al. (2014)*, by injecting 5 µl onto an Astec Cyclobond I2000 chiral HPLC column (250 mm ×4.6 mm; 5 µm particle size) (Sigma Aldrich) with an injection loop size of 2 µL. The column oven temperature was controlled and maintained at 35°C throughout the acquisition and the mobile phases were as follows: 50 mM aqueous ammonium acetate (eluent A) and 90:10 (% v/v) acetonitrile:purified water (eluent B). The mobile phase flow rate was maintained at 600 µL/min and was introduced directly into the mass spectrometer with no split. The mobile phase gradient profile was as follows: Starting condition 100% eluent B, 0.0–1.0 min: 100% B to 25% B, 1.0–22.0 min: 25% B, 22.0–22.5 min: 25% B to 0% B, 22.5–23.0 min: 0% B, 23.0–24.0 min: 0% B to 100% B, 24.0–30.0 min: 100% B. The mass spectrometer was controlled by MS Workstation 8.2.1 software (Bruker Daltonics) using electrospray ionization operated in negative ion mode. The following parameters were used to acquire Multiple Reaction Monitoring (MRM) data: spray voltage: 3.0 kV, cone temperature: 350°C, cone gas flow 20, probe gas flow: 50, nebulizer gas flow: 50, heated probe temperature: 350°C, exhaust gas: on, CID: 1.5 mTorr. The MRM scan time was set to 1000 ms for DMAPP and IPP, and 200 ms for MVA-P with standard resolution for all transitions. The collision energy (CE) was optimised for each transition. The quantifier was $m/z$ 245.0 → 79 (CE: 16 eV) and qualifier $m/z$ 245.0→ 159 (CE: 16 eV) for both DMAPP and IPP. For the internal standard MVA-P the quantifier was $m/z$ 227.0 → 79 (CE: 24 eV) and qualifier $m/z$ 227.0→ 97 (CE: 13 eV). Initial retention times (RT) were 14.1 min (MVA-P) 19.2 min (DMAPP) and 23.6 min (IPP) but shifted to less retention as the column presumably deteriorated during the runs. For quality control (QC) and to ensure correct peak integration a 1 µM standard DMAPP/IPP mix was injected every 12[th] sample. The RTs decreased in a linear fashion from the first 1 µM QC standard to the last QC standard (n = 52) with 0.024 min, 0.044 min, and 0.061 min per injection for MVA-P, DMAPP, and IPP respectively ($R^2$ = 0.990, $R^2$ = 0.991, $R^2$ = 0.989). Analytes were integrated manually.

To obtain quantitative data, a matrix-matched internal standard calibration was used. Analyte stock solutions were prepared in 90% (v/v) acetonitrile and were diluted with blank matrix extract, extracted with 90:10 (% v/v) acetonitrile:Milli-Q water). The internal standard was added to the final HPLC vial at a concentration of 16 µM. The calibration curve ranged from 0.25 µM to 10 µM with $R^2$ values of 0.968 and 0.981 for DMAPP and IPP, respectively. For both calibration curves a $1/x^2$ weighting factor was applied. Sample concentrations lower than the lowest standard were obtained through extrapolation of the calibration curve. The limit of quantification (LOQ) was approximated, using the lowest standard as reference (0.25 µM, n = 4), as 10x the signal-to-noise ratio. The LOQ estimate was 0.033 and 0.045 µM for DMAPP and IPP respectively. The 1 µM QC standard (n = 8) recovery was 85.6 (RSD 18.7%) and 93.2 (RSD 15.9%) for DMAPP and IPP respectively. Additionally, five standards with different DMAPP/IPP ratios were injected to verify the ratio accuracy. DMAPP: IPP ratios fortified were 10, 2, 1, 0.5, and 0.1, while ratios found were 11.1, 1.8, 0.96, 0.56, and 0.10 (bias ranging from −9.9 to 12.5% with a mean bias of 1.9%).

## Protein quantification

Cells were harvested for proteomics analyses at the same time point as metabolomics samples. Cell pellets corresponding to 1 ml of cultures of an optical density of $OD_{600} = 5$ were processed according to *Rennig et al. (2019)*, both regarding preparation of samples, the applied gradient on the CapLC system and the settings for Orbitrap HF_X mass spectrometer. Here, a total of 1 µg of peptides/sample was injected into the mass spectrometer. After acquisition the raw files were analysed using Proteome Discoverer 2.3 (P.D. 2.3) in order to identify and quantify detected proteins. The following software settings were used: Fixed modification: Carbamidomethyl (C) and Variable modifications: oxidation of methionine residues. First search mass tolerance 10 ppm and a MS/MS tolerance of 0.02 Da., trypsin as proteolytic enzyme and allowing two missed cleavages. FDR was set at 0.1%. For match between runs the ΔRT was set to 0.2 min and the minimum peptide length was set to 7. As database for the searches the *E. coli* W proteome (UP000008525) was used combined with a contaminant database (cRAP) and the sequences of heterologous HDRs (see *Table 1*) and lycopene production proteins Idi (Genbank ID AAC32208.1), CrtE (WP026199135.1), CrtI (AAA64981.1), and CrtB (WP020503292.1). Normalization of the data across samples was done with P.D. 2.3. using total peptide amount, meaning all identified peptides in the individual samples are used for normalization, while using one file as master file to which all other counts are normalized. For quantification only unique peptides were used, and for all HDR proteins, hits were manually inspected to ensure correct identification and quantification. HDR overexpression strains were compared by analysing normalized peptide counts using one-way analysis of variance (ANOVA) or Welch's ANOVA test in case of unequal variances, respectively. Where reported, p-values were adjusted for multiple comparison testing using Dunnett's method, n ≥ 3 biological replicates.

## Isoprene production

The different pT-HDR plasmids were transformed into *E. coli* WΔcscR, *lacZ::Pt-DXS*, *arsB::PaISPS* (del2-52,A3T,L70R,S288C). All growth media contained 250 mg $L^{-1}$ ampicillin for plasmid maintenance. Strains were grown in LB medium until stationary phase, then diluted in 0.5 ml TB medium containing 0.1 mM IPTG to a starting $OD_{600}$ of 0.1, and grown at 30°C, with 250 rpm shaking. Cultures were grown in 20 ml sealed gas chromatography vials and isoprene was quantified after 48 hr as described previously (*Vickers et al., 2015*).

## Sequence alignments and generation of phylogenetic trees

HDR protein sequences were downloaded from the results of a BLASTP search with *A. thaliana* HDR against land plants (taxid: 3193), manually removing identical duplicates and obvious pseudogenes (deletions or mutations in highly conserved regions). Sequences were truncated to remove N-terminal chloroplast targeting sequences and aligned using CLC Main Workbench (Qiagen). HDR phylogenetic tree (unrooted) was generated using maximum likelihood phylogeny, neighbour-joining method, WAG protein substitution model, and bootstrap analysis with 100 replicates, also in CLC Main Workbench. Phylogenetic trees were visualised using Interactive Tree of Life (iTOL) v3 (*Letunic and Bork, 2016*).

## Acknowledgements

Research at the University of Queensland was funded by an Australian Research Council grant to LKN and CEV. Research at the Center for Biosustainability was supported by The Novo Nordisk Foundation under NFF grant number NNF10CC1016517. JPG was supported by a Marie Curie International outgoing Fellowship within the 7th European Community Framework Programme. CEV was supported by Queensland Government Smart Futures and Accelerate Fellowships. Metabolomics Australia is part of the Bioplatforms Australia network, funded through the Australian Government's National Collaborative Research Infrastructure Strategy (NCRIS). The authors would like to thank James Behrendorff for valuable feedback on this work.

## Additional information

### Funding

| Funder | Grant reference number | Author |
| --- | --- | --- |
| Australian Research Council | DP140103514 | Lars K Nielsen<br>Claudia E Vickers |
| Novo Nordisk Foundation | NNF10CC1016517 | Lars Schrübbers<br>Tune Wulff<br>Morten O A Sommer<br>Lars K Nielsen<br>Mareike Bongers |
| Marie Skłodowska-Curie Actions | FP7-PEOPLE-2013-IOF. Project: 623679 | Jordi Perez-Gil |
| Department of Education, Australian Government | National Collaborative Research Infrastructure Strategy (NCRIS) | Mark P Hodson<br>Lars K Nielsen |
| Queensland Government | Accelerate Fellowship | Claudia E Vickers |

The funders had no role in study design, data collection and interpretation, or the decision to submit the work for publication.

### Author contributions

Mareike Bongers, Conceptualization, Data curation, Investigation, Visualization, Methodology, Project administration; Jordi Perez-Gil, Investigation, Methodology; Mark P Hodson, Lars Schrübbers, Data curation, Methodology; Tune Wulff, Formal analysis, Performed and analysed the proteomics work; Morten OA Sommer, Resources, Funding acquisition; Lars K Nielsen, Conceptualization, Resources, Supervision, Funding acquisition; Claudia E Vickers, Conceptualization, Supervision, Funding acquisition, Investigation

### Author ORCIDs

Mareike Bongers https://orcid.org/0000-0003-4739-3852
Jordi Perez-Gil https://orcid.org/0000-0002-5632-9556
Mark P Hodson https://orcid.org/0000-0002-5436-1886
Tune Wulff https://orcid.org/0000-0002-8822-1048
Lars K Nielsen https://orcid.org/0000-0001-8191-3511
Claudia E Vickers https://orcid.org/0000-0002-0792-050X

### Decision letter and Author response

Decision letter https://doi.org/10.7554/eLife.48685.sa1
Author response https://doi.org/10.7554/eLife.48685.sa2

## Additional files

### Supplementary files

- Transparent reporting form

## Data availability

All data generated or analysed during this study are included in the manuscript and supporting files. Source data files have been provided for Figures 2 and 3.

The following datasets were generated:

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
