## [Decision Letter]

[Editors’ note: the authors submitted for reconsideration following the decision after peer review. What follows is the decision letter after the first round of review.]

Thank you for submitting your work entitled "Adaptation of hydroxymethylbutenyl diphosphate reductase enables volatile isoprenoid production" for consideration by *eLife*. Your article has been reviewed by two peer reviewers, and the evaluation has been overseen by a Reviewing Editor and a Senior Editor. The reviewers have opted to remain anonymous.

Based on the reviews below, we regret to inform you that your work will not be considered further for publication in *eLife*. While both reviewers found the work very interesting, they also expressed some major concerns. Reviewer 1 highlights a number of factors that would need to be better controlled to support the authors conclusions. To overcome these concerns additional experiments would be required.

Reviewer #1:

The architecture of the isoprenoid pathway requires fusion of two different C5 intermediates, DMADP and IDP, to produce all but the simplest products. Since the formation of the growing prenyl diphosphate chains is initiated with the allylic substrate DMADP and followed by multiple additions of the homo-allylic IDP, different ratios of DMADP:IDP are needed for different length isoprenoid products. Thus the ratio of DMADP:IDP available could control the rate of formation of different isoprenoid classes.

The authors of this contribution have discovered that for C5 intermediates derived from the MEP pathway, the ratio of DMADP:IDP is not fixed at 1:5 by the last enzyme of the pathway (hydroxymethylbutenyl diphosphate reductase- HDR) as once believed. Different HDR enzymes in different species seem to make a range of ratios, which might be important in controlling the formation of various isoprenoids, especially the gaseous isoprene, which is a product of DMADP solely. The finding of two apparent HDR paralogs in some gymnosperms is also fascinating as it suggests that plants might regulate DMADP:IDP ratios in different ways in distinct cell types depending on the varying spectrum of isoprenoids needed. This is an exciting story, but I have two concerns with the results reported.

1) The major concern is that the authors have not really measured the direct product ratio of the enzyme in a standard way by characterizing the isolated enzyme in vitro. Instead they have measured the concentrations of DMADP and IDP in living *E. coli* cells in which the enzymes have been over-expressed. This is partly understandable as HDR, like other Fe-S cluster proteins, is oxygen-sensitive and considered hard to assay in vitro. However, many other processes affect the in vivo concentrations of DMADP and IDP in living cells, and so the authors might do a better job of controlling these factors to make a stronger case that the differences they see in the HDRs of various species are real.

The authors have over-expressed these HDRs in cells with over-expressed DXS ("to achieve intracellular IDP and DMADP concentrations above the quantification limits", DXS is supposed to be the rate-controlling step of the pathway), and with a new IDI enzyme and an additional biosynthetic pathway for the carotenoid lycopene (apparently to provide a sink for all of the excess IDP and DMADP, which are toxic). The result seems a very artificial, high flux environment for any isoprenoid synthesizing process, and it is hard to know if the ratios measured really reflect the intrinsic properties of the HDR, particularly since the expressed HDR is always mixed with the *E. coli* HDR.

Each HDR was expressed in the same cassette with the same promoter, but the ratios could vary depending on the level of expression of the *E. coli* enzyme vs. the heterologous HDR. Can the authors verify (by proteomics) that the protein levels of the tested HDRs in each experiment were the same? This might help show that the in vivo pools reflect real differences among the enzymes. One might expect that the host *E. coli* HDR responds in a varying way to the expression of heterologous enzymes, and so its level should be measured also. Hopefully, its contribution is also constant.

Another approach might be to down-regulate the *E. coli* HDR when the foreign HDR to be tested is over-expressed. This might avoid the toxic effects, and allow the foreign HDR to be tested with less competition.

The authors might additionally determine not only the ratio of DMADP to IDP, but also the absolute amounts of the two compounds to see if they are in the range expected in a cell, considering all of the genetic manipulations being carried out.

I realise that all of these suggestions require additional work. Perhaps there is another way to validate such in vivo measurements as representing the real characteristics of the enzyme when tested in vitro. Maybe an attempt could be made to try expression, purification and in vitro assay of say two of the HDRs from opposite ends of the DMADP:IDP scale.

2) One of the authors' most impressive achievements was to develop an HPLC separation scheme for IDP and DMADP. But as this procedure has never been formally published in the refereed literature, the authors should provide chromatograms with accompanying mass spectra to verify their separation. Also, the previously published retention times vary greatly from those reported here (subsection “Quantification of IPP and DMAPP”), which seems to deserve some comment.

Reviewer #2:

This manuscript describes mechanistically interesting novel data on the important role of hydroxymethylbutenyl diphosphate reductase (HDR/IspH) regulating the ratio of DMAPP and IPP as prerequisite for enhanced metabolic flux to isoprene (and monoterpenes) or long-chain terpenoids such as lycopene.

The authors have chosen an elegant molecular biological approach for their investigations: In addition to HDR genes from *E. coli* and Synechococcus, they investigated 8 HDR genes from plant species of originating from different taxa of the plant. In a first step, functional HDR genes were identified by complementing lethal knockout mutants of HDR/IspH in *E. coli*. It was found that the overexpression of some HDR genes led to growth inhibition in *E. coli*, which was attributed to the toxicity of accumulating prenyl phosphates.

For the main investigations, the *E. coli* metabolism was modified in which an overexpressed DXR gene led to an increased metabolic flow of the MEP pathway; a lycopene synthase or an isoprene synthase were incorporated to produce a metabolic sink to higher terpenoids (C40) and volatile (C5) hemiterpenes. In addition, a heterologous IPP isomerase (IDI) was incorporated to ensure sufficient IPP for the biosynthesis of longer chain terpenoids.

DMAPP and IPP were measured with directed LC-MS/MS which is state-of-the-art.

It is impressive to see how different the DMAPP/IPP ratios are adjusted by the different HDR enzymes. Enzymes from plants with naturally high isoprene emission have very high DMAPP/IPP ratios. Enzymes from organisms without isoprene emission show very low ratios, similar to HDR enzymes from gymnosperms.

Very interesting is the result shown in Figure S4 that the DMAPP/IPP ratios in vivo depend on the end product of the metabolism: higher at isoprene emission, lower at lycopene accumulation. I would suggest to show this result in the main text. I would also like to see a more detailed discussion of this observation. A point that is also a bit neglected in the discussion is the role of IDI in the fine tuning of DMAPP/IPP ratios. As rightly noted, the DMAPP/IPP ratio measured in vivo does not correspond to the (not measured) in vitro ratio, but is changed by the activity of the IDI. For isoprene emitting oaks it was shown that the activity of the IDI correlates positively with ISPS and isoprene emission (Brüggemann and Schnitzler, 2002), which indicates a rate-limiting role of the IPI as mentioned.

To understand the fine tuning of the DMAPP/IPP ratio between HDR/IspH, IDI and the sinks towards the different soluble and volatile end products additional information on IDI is necessary: i.e. what is the in vitro (in vivo) ratio of the IDI used in the present experiments? I'm wondering why no experiments without overexpressed (and knocked out) IDI have been performed. How does the absence of this enzyme (absence of overexpression) influence the in vivo ratios of DMAPP/IPP in the different systems (lycopene vs isoprene)? I hope the authors can provide this information from experiments not yet shown.

Overall a very interesting novel work demonstrating the potential role of HDR/IspH to control the ratio of C5 terpenoid intermediates central for the channeling of metabolites to the higher terpenoids/isoprenoids and or volatile compounds, such as isoprene and monoterpenes. Beside understanding the metabolic regulation within the MEP pathway in natural systems knowledge on the ratio of the enzyme products of HDR/IspH also will provide the avenue for much more efficient biotechnological system for all type of terpenoids.

[Editors’ note: further revisions were suggested prior to acceptance, as described below.]

Thank you for submitting your revised article "Adaptation of hydroxymethylbutenyl diphosphate reductase enables volatile isoprenoid production" for consideration by *eLife*.

Your article has been reviewed by one of the two reviewers who have seen the original submission. The second reviewer was unfortunately no longer available to assess the revised paper. The evaluation has been overseen by a Reviewing Editor, Joerg Bohlmann, and Ian Baldwin as the Senior Editor. The reviewer has opted to remain anonymous.

We find that paper has been substantially improved, but there are a few remaining concerns in the external reviewer's report.

Reviewer #1:

The authors have made extensive revisions and significantly improved their manuscript. In particular, the new proteomics measurements help to assure that the activity of the over-expressed HDR proteins is not affected by the native *E. coli* protein or the expression of the genes on the lycopene plasmid. Obtaining in vitro assay data on these proteins would clearly be the "gold standard" for comparison among them, but I can accept that this is just not yet technically attainable, and they have done what they can to increase the accuracy of their in vivo measurements. I also appreciated the chromatograms to verify the DMADP-IDP separation.

Two minor concerns that could be addressed:

1) Compared to the first submission, the DMADP:IDP ratio for *Prunus persica* has now decreased by 10 fold, while the ratios for other species have declined by more than 50%. Since these are the core data of the paper, such big changes do not inspire confidence in the analytical methods. Is there a reason for such variance from the previous version, such as a change in the analysis protocol or a change in calibrating factors?

2) In subsection “Isoprene production” it states that "isoprene was quantified as described previously", but the reference given is incorrect. Please correct this.

---

## [Author Response]

[Editors’ note: The authors appealed the original decision. What follows is the authors’ response to the first round of review.]

Reviewer #1:The architecture of the isoprenoid pathway requires fusion of two different C5 intermediates, DMADP and IDP, to produce all but the simplest products. Since the formation of the growing prenyl diphosphate chains is initiated with the allylic substrate DMADP and followed by multiple additions of the homo-allylic IDP, different ratios of DMADP:IDP are needed for different length isoprenoid products. Thus the ratio of DMADP:IDP available could control the rate of formation of different isoprenoid classes.The authors of this contribution have discovered that for C5 intermediates derived from the MEP pathway, the ratio of DMADP:IDP is not fixed at 1:5 by the last enzyme of the pathway (hydroxymethylbutenyl diphosphate reductase- HDR) as once believed. Different HDR enzymes in different species seem to make a range of ratios, which might be important in controlling the formation of various isoprenoids, especially the gaseous isoprene, which is a product of DMADP solely. The finding of two apparent HDR paralogs in some gymnosperms is also fascinating as it suggests that plants might regulate DMADP:IDP ratios in different ways in distinct cell types depending on the varying spectrum of isoprenoids needed. This is an exciting story, but I have two concerns with the results reported.1) The major concern is that the authors have not really measured the direct product ratio of the enzyme in a standard way by characterizing the isolated enzyme in vitro. Instead they have measured the concentrations of DMADP and IDP in living *E. coli* cells in which the enzymes have been over-expressed. This is partly understandable as HDR, like other Fe-S cluster proteins, is oxygen-sensitive and considered hard to assay in vitro. However, many other processes affect the in vivo concentrations of DMADP and IDP in living cells, and so the authors might do a better job of controlling these factors to make a stronger case that the differences they see in the HDRs of various species are real.The authors have over-expressed these HDRs in cells with over-expressed DXS ("to achieve intracellular IDP and DMADP concentrations above the quantification limits", DXS is supposed to be the rate-controlling step of the pathway), and with a new IDI enzyme and an additional biosynthetic pathway for the carotenoid lycopene (apparently to provide a sink for all of the excess IDP and DMADP, which are toxic). The result seems a very artificial, high flux environment for any isoprenoid synthesizing process, and it is hard to know if the ratios measured really reflect the intrinsic properties of the HDR, particularly since the expressed HDR is always mixed with the *E. coli* HDR.

This is a valid argument that we have now addressed with additional experiments. Our initial experimental setup was guided mainly by two factors:

a) being able to directly compare a relatively large number of HDR enzymes to each other, which precludes in vitro characterization of these difficult-to-express proteins, or expression and analysis in plants

and

b) using *E. coli*, which has a very low native flux through the MEP pathway, low MEP pathway gene expression levels and no apparent transcriptional or other regulation of MEP pathway genes in response to pathway engineering (Bongers et al., 2015). In this previous study, we established that MEP pathway gene expression in *E. coli* is not affected by expression of the lycopene pathway plasmid (including idi). With this in mind, our *E. coli*-based experimental system provides a much ‘cleaner’ background to study differences between plant HDRs compared to their native hosts.

In this revised submission, we have used LC-MS/MS proteomics to quantify proteins that could influence the in vivo ratio of DMAPP:IPP (Figure 2—figure supplement 2). We have confirmed that:

· native HDR protein abundance is the same across all strains including the negative control (except for the *E. coli* HDR overexpression strain, as intended), demonstrating that there is no regulation of *E. coli* HDR expression in response to overexpressing the heterologous HDRs.

· The same is true for the native *E. coli* Idi, as well as CrtE, CrtB, CrtI and Hp_Idi_, expressed from the lycopene production plasmid. These data demonstrate that if these proteins affect the DMAPP:IPP ratio measured in our setup, their influence is constant across all strains.

· *E. coli* HDR peptide counts increase >50-fold in the EcHDR overexpression strain compared to all others. In the absence of a dedicated standard curve for the quantified peptides, counts cannot be assumed to increase linearly with protein abundance; however, these data strongly indicate a dramatic increase in HDR abundance upon overexpression compared to the baseline. The heterologous HDRs from other species (though not directly comparable, see comments below) show similarly strong expression levels as the overexpressed EcHDR. This further strengthens the view that the low and constant native EcHDR expression plays a minor role in determining the observed DMAPP:IPP ratios.

Expression of high DMAPP-producing HDRs causes severe toxicity in *E. coli*, making the co-overexpression of a metabolic sink (and, in some cases, idi) necessary to prevent cell death (Figure 2—figure supplement 1B). No growth defect was caused by overexpression of the native *E. coli* HDR, or by any HDRs that favour the production of IPP over DMAPP. This further supports the argument that the observed differences in DMAPP:IPP ratio cannot be sufficiently adjusted by *E. coli*’s native metabolism or available Idi activity.

We have now quantified both IPP and DMAPP in the manuscript (Figure 2—figure supplement 3), and calculated the intracellular concentrations of these compounds (using an accepted method for calculating cell count and volume based on culture density: Volkmer 2011, PloS One). The generated intracellular concentrations of DMAPP and IPP range between 0.5 – 8.7 μM, which may be high for *E. coli*, but certainly seems within physiologically relevant boundaries.

Each HDR was expressed in the same cassette with the same promoter, but the ratios could vary depending on the level of expression of the *E. coli* enzyme vs. the heterologous HDR. Can the authors verify (by proteomics) that the protein levels of the tested HDRs in each experiment were the same? This might help show that the in vivo pools reflect real differences among the enzymes.

Direct quantitative comparison between HDR expression levels is technically challenging because there are no proteotypic peptides that are shared between all HDRs in this set. The only way we could conceive to directly compare different HDR protein levels would be to use protein fusion tags such as His- or FLAG-tag and quantify based on those. We discarded this option because fusion tags may influence protein activity, and because (re-) cloning of high-DMAPP producing HDRs is very time-consuming due to toxicity issues. We now present LC-MS/MS-based proteomics data for all tested HDRs (Figure 2—figure supplement 2C), showing that all are highly expressed on the basis of total normalized counts and the number of peptides detected (between 28-52 peptides detected for each HDR). Using this method, it is not valid to compare counts for non-identical peptides because their different physicochemical properties affect behavior in the mass spectrometer, making results not quantitatively comparable. Since there is no single peptide shared by all tested HDRs, the presented data can only be seen as approximate indications of protein abundance. However, the data suggests that all HDRs are strongly overexpressed, and Pearson’s correlation analysis showed that there is no correlation between HDR ‘abundance’ and DMAPP:IPP ratio. This is now also explained in the main text.

One might expect that the host *E. coli* HDR responds in a varying way to the expression of heterologous enzymes, and so its level should be measured also. Hopefully, its contribution is also constant.

See above: EcHDR expression is constant, and relatively low unless it is specifically overexpressed.

Another approach might be to down-regulate the *E. coli* HDR when the foreign HDR to be tested is over-expressed. This might avoid the toxic effects, and allow the foreign HDR to be tested with less competition.

See above: *E. coli* HDR is expressed at low levels, and overexpressing this gene does not cause any toxicity. The toxicity is only linked to high DMAPP-producing HDR enzymes. The argument that *Ec*HDR has a limited influence in our experimental setup is further supported by Figure 2C, showing that overexpression of *Ec*HDR shifts the ratio slightly towards more IPP compared to the ‘no HDR overexpression’ control. This is in agreement with all previous studies showing a 1:5 DMAPP to IPP ratio for the *E. coli* enzyme.

The authors might additionally determine not only the ratio of DMADP to IDP, but also the absolute amounts of the two compounds to see if they are in the range expected in a cell, considering all of the genetic manipulations being carried out.

Without enhancing MEP pathway flux through engineering, we have found that metabolite measurements (isoprene and lycopene, as well as MEP pathway intermediates) are at or below the detection limit with mass spectrometry-based analytic methods.

Compared to plants, which can redirect up to 20% of recently fixed carbon into isoprenoids and produce many essential isoprenoid compounds, *E. coli* has minimal requirements for isoprenoids and naturally very low MEP pathway flux. Indeed, two decades of metabolic engineering of the MEP pathway have demonstrated its reluctance to sustain high flux and it remains challenging to produce high yields of isoprenoid compounds using this pathway in *E. coli* (DOI: 10.1093/femsle/fny079). The high demand for redox equivalents, which in plants is met by electrons coming from photosynthesis, may be one of the reasons for this. Hence, overexpressing *dxs* (as done in our study) is only the first step of several required to engineer high flux through the MEP pathway in *E. coli*. On this basis, and with our new observations that intracellular concentrations are in the low μM range, we do not think that the metabolite levels generated in this work are outside of physiologically relevant ranges.

I realise that all of these suggestions require additional work. Perhaps there is another way to validate such in vivo measurements as representing the real characteristics of the enzyme when tested in vitro. Maybe an attempt could be made to try expression, purification and in vitro assay of say two of the HDRs from opposite ends of the DMADP:IDP scale.

We feel that demonstrating the differences in HDR production ratios is most relevant in an in vivo setting with realistic reaction conditions and redox partners, where their effect on downstream isoprene production can be directly observed. Achieving suitable in vitro reaction conditions for oxygen-sensitive FeS-cluster HDRs, as well as finding a physiologically relevant redox donor is documented to be very challenging (DOI: 10.1021/ja903778d), and requires specialist facilities which we do not currently have access to. We acknowledge that data from in vitro assays of selected HDRs could be valuable in future studies, but for this study we feel that we have sufficiently improved our in vivo analysis and interpretation.

Reviewer #2:[…]Very interesting is the result shown in Figure S4 that the DMAPP/IPP ratios in vivo depend on the end product of the metabolism: higher at isoprene emission, lower at lycopene accumulation. I would suggest to show this result in the main text. I would also like to see a more detailed discussion of this observation.

This result might indeed be better presented in the main text, as it addresses the question raised by both reviewers of how robust and context-independent the observed differences in HDRs are. We have now included this data in Figure 2C, and discuss it in the main text (Results and Discussion paragraph four). We show that when using a different metabolic sink (isoprene synthase instead of lycopene), the relative differences between at least a subset of the tested HDRs, stay constant. Even though the absolute ratio of DMAPP:IPP shifts depending on the downstream sink, differences between the tested HDRs can be captured with both experimental setups.

A point that is also a bit neglected in the discussion is the role of IDI in the fine tuning of DMAPP/IPP ratios. As rightly noted, the DMAPP/IPP ratio measured in vivo does not correspond to the (not measured) in vitro ratio, but is changed by the activity of the IDI. For isoprene emitting oaks it was shown that the activity of the IDI correlates positively with ISPS and isoprene emission (Brüggemann and Schnitzler, 2002), which indicates a rate-limiting role of the IPI as mentioned.To understand the fine tuning of the DMAPP/IPP ratio between HDR/IspH, IDI and the sinks towards the different soluble and volatile end products additional information on IDI is necessary: i.e. what is the in vitro (in vivo) ratio of the IDI used in the present experiments? I'm wondering why no experiments without overexpressed (and knocked out) IDI have been performed. How does the absence of this enzyme (absence of overexpression) influence the in vivo ratios of DMAPP/IPP in the different systems (lycopene vs isoprene)? I hope the authors can provide this information from experiments not yet shown.

We found that IDI is absolutely required when studying any of the high DMAPP-producing HDR enzymes in *E. coli* (Figure 2F, and Figure 2—figure supplement 1B, and data not shown). Any attempts to clone, express or analyze these high DMAPP HDRs in the absence of IDI have resulted in mutated, nonfunctional HDR genes. We agree that the influence of IDI on the DMAPP:IPP ratio is an important question that warrants further investigation, particularly in vivo in plants which likely use regulation of IDI expression as a further mechanism to control isoprenoid production.

In the present study, the important question is whether IDI co-expression artifactually influences our results. In our revised submission we have validated that IDI expression is constant across all strains (Figure 2—figure supplement 2A and 2D). Therefore, the different HDR enzymes must be the primary driver of the 40-fold differences in in vivo DMAPP:IPP ratios between different HDRs. While the IDI enzyme must affect DMAPP and IPP concentrations to some extent, it does not appear to have a major impact on our observations.

Furthermore, our proteomics data suggest that IDI expression is overall very low, based on both the total counts, number of peptides detected (1 peptide for Ec_Idi_, 3 peptides for Hp_Idi_, compared to 28-50 peptides detected for each HDR protein) and the normalized rank (Ec_Idi_ ranks at around 2300 out of 2500 quantified proteins in our samples). These data have to be seen in light of the inherent limitations of non-targeted proteomics, yet they strongly suggest low relative protein abundance for both IDIs present in our system.

[Editors’ note: what follows is the authors’ response to the second round of review.]

We find that paper has been substantially improved, but there are a few remaining concerns in the external reviewer's report.Reviewer #1:The authors have made extensive revisions and significantly improved their manuscript. In particular, the new proteomics measurements help to assure that the activity of the over-expressed HDR proteins is not affected by the native *E. coli* protein or the expression of the genes on the lycopene plasmid. Obtaining in vitro assay data on these proteins would clearly be the "gold standard" for comparison among them, but I can accept that this is just not yet technically attainable, and they have done what they can to increase the accuracy of their in vivo measurements. I also appreciated the chromatograms to verify the DMADP-IDP separation.Two minor concerns that could be addressed:1) Compared to the first submission, the DMADP:IDP ratio for Prunus persica has now decreased by 10 fold, while the ratios for other species have declined by more than 50%. Since these are the core data of the paper, such big changes do not inspire confidence in the analytical methods. Is there a reason for such variance from the previous version, such as a change in the analysis protocol or a change in calibrating factors?

The reviewer’s observations are correct, but the reason for the difference is that we have substantially improved the analytical method. It is important to note that the ranked order of HDR proteins has not changed in the new data set (ranked in terms of their DMAPP:IPP ratios), and the improvements to the analytical method give us confidence that the DMAPP and IPP quantification is more accurate.

In our earlier dataset we reported DMAPP and IPP as a ratio of their LC-MS/MS peak areas. The accuracy of these ratios was limited by the fact that we had not managed to achieve full baseline separation of DMAPP and IPP. The to some extent overlapping and tailing peaks particularly affected peak area calculations in cases with very high DMAPP (first eluting analyte) or very low IPP (for example, the IPP concentration in *P. persica* samples was very low).

We subsequently made substantial improvements to the analytical method. We can now report a method that results in complete baseline separation of DMAPP and IPP peaks and an appropriate internal standard. With this improved method we can now report actual concentrations for DMAPP and IPP, and calculate DMAPP:IPP ratios with much greater accuracy.

Since this is the first publication of a method for LC-MS/MS separation and absolute quantification of DMAPP and IPP, we have taken great care to describe this technically challenging method in detail, including QC steps and potential pitfalls, to ensure reproducibility in other labs.

2) In subsection “Isoprene production” it states that "isoprene was quantified as described previously", but the reference given is incorrect. Please correct this.

This reference has been corrected now.